# Impaired Heat Shock Protein Expression in Activated T Cells in B-Cell Lymphoma

**DOI:** 10.3390/biomedicines10112747

**Published:** 2022-10-28

**Authors:** Zarema Albakova, Yana Mangasarova, Alexander Sapozhnikov

**Affiliations:** 1Department of Biology, Lomonosov Moscow State University, Moscow 119192, Russia; 2Chokan Limited Liability Partnership (LLP), Almaty 050039, Kazakhstan; 3National Medical Research Center for Hematology, Moscow 125167, Russia; 4Department of Immunology, Shemyakin and Ovchinnikov Institute of Bioorganic Chemistry of the Russian Academy of Sciences, Moscow 117997, Russia

**Keywords:** heat shock proteins, HSP90, HSP70, HSP60, extracellular HSPs, T cells, lymphoma

## Abstract

Heat shock proteins (HSPs) are molecular chaperones that act in a variety of cellular processes, ensuring protein homeostasis and integrity. HSPs play critical roles in the modulation of various immune cells. However, the role of HSPs in T cell activation is largely unknown. We show that HSPs are upregulated following CD3/CD28 stimulation, suggesting that HSP expression might be regulated via TCR. We found that B-cell lymphoma (BCL) patients have dysregulated expression of intracellular and extracellular HSPs, immune checkpoints PD-1, CTLA-4, and STAT3 in CD3/CD28-activated T cells. Consistent with previous findings, we show that HSP90 inhibition downregulated CD4 and CD8 surface markers in healthy controls and BCL patients. HSP90 inhibition alone or in combination with PD-1 or CTLA-4 inhibitors differentially affected CD4+ and CD8+ T cell degranulation responses when stimulated with allogeneic DCs or CD3/CD28 in BCL patients. Additionally, we showed that HSP90 inhibition does not significantly affect intracellular PD-1 and CTLA-4 expression in CD3/CD28-activated T cells. These findings may provide the basis for the discovery of novel immunological targets for the treatment of cancer patients and improve our understanding of HSP functions in immune cells.

## 1. Introduction

Heat shock proteins (HSPs) are a family of molecular chaperones contributing to the protein homeostasis and survival of the cell [1]. HSPs are critical for the folding, unfolding, and maturation of client proteins [2,3,4]. HSPs are grouped into distinct families, such as HSP90, HSP70, HSP40, HSP110, HSPB, and chaperonins [4]. HSPs are located in different cellular compartments (cytosol, endoplasmic reticulum (ER), lysosome, mitochondria, nucleus), where they play important roles in various stress response pathways [5,6,7]. HSPs are involved in the regulation of various cellular processes, including unfolded protein responses, mitochondrial metabolism, apoptosis, autophagy, and innate and adaptive immune responses [8,9,10,11,12]. Originally described as intracellular proteins, several studies reported the presence of HSPs in cancer patients’ plasma and serum, extracellular vesicles, and tumor cell surfaces, collectively known as extracellular HSPs (eHSPs) [13,14,15,16,17,18,19,20,21,22]. Surface expression of HSPs often correlates with tumor aggressiveness, increased cell proliferation, and unfavorable clinical outcome [14,23,24,25]. eHSPs have been shown to be implicated in various hallmarks of cancer, including epithelial-mesenchymal transition, angiogenesis, migration, invasion, and tumor immunity [8,26,27].

B-cell lymphomas (BCL) represent a large group of non-Hodgkin’s lymphomas, which can be further divided into indolent (slow-growing) and aggressive (fast-growing) lymphomas [28,29]. The most common indolent lymphoma is follicular lymphoma (FL), whereas the most common aggressive lymphoma is diffuse large B-cell lymphoma (DLBCL) [30]. Several signaling cascades are dysregulated in BCL, including Janus kinase (JAK)-signal transducer and activator of transcription (STAT) (JAK-STAT), nuclear factor-kappa B (NF-κB), B-cell receptor signaling, phosphatidylinositol 3-kinase (PI3K)-serine/threonine protein kinase (AKT) pathway (PI3K/AKT), and mitogen-activated protein kinase (MAPK)/ERK [29,31,32,33]. Current treatments for BCL patients include either chemotherapy combined with anti-CD20 or PD-1 inhibitors [34,35].

Programmed death 1 (PD-1) and cytotoxic T-lymphocyte associated protein-4 (CTLA-4) are immune checkpoints, which negatively regulate T cell function [36]. STAT3 is a transcription factor that is upregulated downstream of various cytokine receptors, playing a critical role in the regulation of CD4^+^ and CD8^+^ T cells [37]. Recently, Mbofung et al. have demonstrated that HSP90 inhibition potentiates the response to PD-1 and CTLA-4 treatment, while Zavareh et al. reported that HSP90 inhibition downregulates PD-L1 surface expression [38,39]. In our previous study, we have shown that the PD-1 inhibitor Nivolumab affects the expression of HSPs in peripheral blood and bone marrow lymphocytes of BCL patients. However, the exact mechanism by which HSPs may modulate immune checkpoints remains unclear [40]. The aim of this study was to examine the expression of intracellular and extracellular HSPs during T cell activation and the effect of combined blockade of HSP90 and immune checkpoints on T cell functional activity in vitro.

Here we show that HSPs are upregulated in T cells following CD3/CD28 stimulation, suggesting that HSP expression may be regulated via TCR signaling. We also show that healthy controls and BCL patients differ in the level of intracellular and extracellular HSP expression in CD3/CD28-stimulated T cells. We further show that HSP90 inhibitors in combination with PD-1 or CTLA-4 inhibitors differently affect CD4^+^ and CD8^+^ T cell degranulation responses following stimulation with CD3/CD28 and allogeneic DCs in BCL patients. Finally, we show that the HSP90 inhibitor does not significantly affect intracellular expression of immune checkpoints (PD-1 and CTLA-4) in CD3/CD28-stimulated T cells.

## 2. Materials and Methods

### 2.1. Samples

Blood samples were collected into EDTA tubes from patients with indolent lymphoma (follicular lymphoma, *n* = 4), aggressive lymphoma (diffuse large B-cell lymphoma, primary mediastinal large B-cell lymphoma, *n* = 5) and healthy controls (*n* = 4). The study involved only newly diagnosed patients with no previous history of treatment. Peripheral blood mononuclear cells were obtained by density gradient centrifugation. The study was approved by the National Medical Research Center for Hematology (Moscow, Russia). All subjects had provided written informed consent in accordance with the Declaration of Helsinki.

### 2.2. T-Cell Activation

T cells were isolated using the Pan T cell isolation kit (Miltenyi Biotec, Germany) and activated for 48 h using T cell TransAct (titer 1:100, Miltenyi Biotec), according to the manufacturer’s instructions. Cells were incubated in RPMI 1640 with L-glutamine, 10% FBS, penicillin/streptomycin (Capricorn Scientific, USA), and human IL-2 (60 IU/mL, Miltenyi Biotec, Germany). For the experiments with a longer activation time, fresh media and hIL-2 were added to the cells at 72 h post-initial activation. Resting T cells were incubated in media alone, without the addition of CD3/CD28 or hIL-2.

### 2.3. Activation Experiment-HSP, PD-1, CTLA-4, and STAT3 Antibodies

T cells were stained with primary antibodies against HSP90α (2G5.G3, Abcam, USA), HSP90β (EPR16621, Abcam, USA), HSP70 (C92F3A-5, Enzo Life Sciences, USA), HSC70 (1B5, Stressgen, USA), HSP40/DNAJB1 (3B9.E6, Abcam, USA), GRP78 (StressMarq Biosciences, USA), and HSP60 (StressMarq Biosciences, USA). Secondary antibodies were goat anti-rabbit IgG H&L (PE) preadsorbed (Abcam), anti-mouse IgG1 (PE) (Miltenyi Biotec, Germany), and anti-rat F(ab’)2 (PE) (Sony Biotechnology, USA). T cells were also stained with anti-human CD279/PD-1 (REA1165, Miltenyi Biotec, Germany) (PE), CD152/CTLA-4 (REA1003, Miltenyi Biotec, Germany) (PE), primary antibody against STAT3 (4G4B45, Sony Biotechnology, USA), and secondary antibody anti-mouse IgG1 (PE) (Miltenyi Biotec, Germany). The FcR blocking reagent (Miltenyi Biotec, Germany) and isotype controls were used to exclude non-specific binding. For intracellular staining, cells were fixed and permeabilized using Cytofix/Cytoperm. Cells were analyzed by flow cytometry.

### 2.4. Mixed Lymphocyte Reaction

For allogeneic mixed lymphocyte reaction (MLR), monocytes were magnetically isolated using CD14 Microbeads (Miltenyi Biotec, Germany) and cultured for 5 days in the presence of IL-4 and GM-CSF (all Miltenyi Biotec, Germany). T cells from BCL patients and healthy controls were incubated with allogeneic mo-DCs in the presence or absence of either 0.1 µM HSP90 inhibitor (geldanamycin, GA) (Abcam, USA), 10 µg/mL PD-1 inhibitor (Nivolumab) (Opdivo, Bristol-Myers Squibb), or a combination of both drugs for 4 days. PD-1 inhibitor (Pembrolizumab) (10 µg/mL, Keytruda, MSD, USA) and IgG4 isotype (10 µg/mL, BioLegend, USA) were used as positive and negative controls, respectively. The CTLA-4 inhibitor (Ipilimumab) (10 µg/mL, Yervoy, Bristol-Myers Squibb, USA) was also used for the comparison.

### 2.5. Degranulation Assay

Monensin (2 µM, Sony Biotechnology, USA) and anti-CD107a APC-Cy7 (Sony Biotechnology, USA) were added to the cells for the last 5 h of incubation. Cells were then harvested and stained for CD3 (Pacific Blue), CD4 (FITC), and CD8 (PE/Cy5) (all Sony Biotechnology, USA) and analyzed by flow cytometry. Since HSP90 inhibition may affect allogeneic mo-DCs, a similar experiment was performed using CD3/CD28 stimulation.

### 2.6. HSP90 Inhibition experiment – PD-1 and CTLA-4 Expression

T cells were incubated with CD3/CD28 and hIL-2 for 48 h in complete RPMI medium at 37 °C and then were incubated with 0.1 µM geldanamycin or DMSO for an additional 24 h. T cells were then stained for intracellular PD-1- and CTLA-4 expression.

### 2.7. Statistical Analysis

GraphPad Prism 9.4.0 was used for statistical analysis. Results are expressed as mean values ± standard error of the mean (SEM). Both parametric and non-parametric analyses, including two-sample *t*-test, ANOVA, and the Mann–Whitney test, were applied. A value of *p* < 0.05 was considered statistically significant.

## 3. Results

### 3.1. Intracellular and Extracellular HSP, PD-1, and CTLA-4 Expression in CD3/CD28-Activated T Cells

Intracellular HSPs were upregulated in T cells during 48 h of CD3/CD28 stimulation in healthy controls, though only HSP90α, HSP90β, and HSC70 reached statistical significance (*p* < 0.05) (Figure 1A). Since immune checkpoints are associated with T cell activation, we also analyzed the expression of PD-1 and CTLA-4 during the anti-CD3/CD28 challenge. As expected, the expression of PD-1 and CTLA-4 was increased, with CTLA-4 reaching statistical significance (Figure 1A). Healthy controls show ~2-fold increase in HSP90β expression compared to patients with indolent lymphoma (Figure 1B). Activated T cells show higher expression of HSC70 compared to resting T cells in patients with indolent lymphoma, whereas higher PD-1 expression was observed in activated T cells compared to resting T cells in patients with aggressive lymphoma (Appendix A). CTLA-4 was also upregulated ~2–2.5-fold in T cells of healthy controls compared to patients with indolent and aggressive lymphoma (Figure 1B). It is interesting to note that there was no statistical difference in extracellular HSPs in resting and activated T cells in healthy controls (Appendix A). By contrast, patients with indolent lymphoma had higher expression of extracellular HSPs compared to healthy controls, though the difference was not statistically significant (Figure 1C, Appendix A). Notably, T cells in patients with aggressive lymphoma had a different pattern of eHSPs compared to T cells in patients with indolent lymphoma (Figure 1C, Appendix A). Since STAT3 is a client of HSP90 and is involved in the regulation of immune checkpoints, we sought to analyze STAT3 expression in activated T cells of lymphoma patients. We show that T cell activation does not significantly upregulate STAT3 expression in T cells of lymphoma patients as compared to healthy controls (Figure 1D). These data suggest that intracellular HSP upregulation is a normal cellular response to T cell activation and that this response is dysregulated in activated T cells in lymphoma patients, which was further supported by the dysregulated expression of immune checkpoints and STAT3 in lymphoma patients. Aberrant expression of extracellular HSPs in CD3/CD28-activated T cells was also associated with lymphoma patients. However, further studies are required to assess the effect of T cell activation on the expression of extracellular HSPs.

### 3.2. HSP90 Inhibition Affects Surface CD4 and CD8 Expression and T-Cell Degranulation Response In Vitro

Since HSP and immune checkpoint expression are associated with T cell activation, we next sought to analyze the effect of HSP90 inhibitors alone or in combination with PD-1 or CTLA-4 inhibitors on T cell degranulation response in lymphoma patients. To study the effects of HSP90 inhibitors and immune checkpoint inhibitors, we used allogeneic mixed lymphocyte reaction (MLR). Since the HSP90 inhibitor may also affect mo-DC responses during MLR, anti-CD3/CD28 stimulation was performed in a parallel experiment. Consistent with previous findings, we found that the HSP90 inhibitor downregulated CD4 and CD8 surface expression on activated T cells in healthy controls and lymphoma patients (Figure 2) [41]. Furthermore, we showed that the HSP90 inhibitor downregulated CD107a expression on CD4^+^T cells in healthy controls during MLR and anti-CD3/CD28 stimulations (Figure 3A,B). By contrast, the HSP90 inhibitor upregulated CD107a expression on the surface of CD4^+^ T cells, and PD-1 and CTLA-4 inhibitors did not abrogate this effect in both allogeneic DC and anti-CD3/CD28 stimulations (Figure 3A,B). Similar to CD4^+^ T cells, HSP90 inhibition downregulated CD107a expression on CD8^+^ T cells in healthy controls, though the difference did not reach statistical significance (Figure 3C,D). CD8^+^ T cells from lymphoma patients show significantly higher surface CD107a expression when treated with the HSP90 inhibitor alone or in combinational treatment with HSP90 inhibitor and PD-1 inhibitors compared to healthy controls (Figure 3C). Healthy controls showed decreased CD107a expression on CD8^+^ T cells when treated with an HSP90 inhibitor in combination with PD-1 inhibitors (Figure 3C). It appears that T cells from lymphoma patients have a higher CD8^+^ T cell degranulation response when stimulated with allogeneic DCs than when stimulated with anti-CD3/CD28 (Figure 3C,D). Furthermore, it appears that different stimulations differentially affect CD8^+^ T cell degranulation responses during combinational treatment (Figure 3C,D).

### 3.3. HSP90 Inhibition and Intracellular PD-1 and CTLA-4

We next sought to analyze the effect of HSP90 inhibition on intracellular PD-1 and CTLA-4 expression. HSP90 inhibition resulted in a slight decrease in PD-1 and a slight increase in CTLA-4 expression (*p* > 0.05) (Figure 4).

## 4. Discussion

HSPs play important roles in folding, degradation, and maturation of client proteins [5,6]. Dysregulated expression of various HSP members has been reported in various tumors, where high HSP expression is typically associated with an unfavorable clinical outcome [14,40,42,43,44]. Several studies have shown that HSP90 inhibitors downregulate surface PD-L1 expression and may also potentiate the effect of PD-1 and CTLA-4 treatments [39]. In our previous study, we found that the PD-1 inhibitor (Nivolumab) affects HSP90 expression in peripheral blood and bone marrow lymphocytes in refractory or recurrent Hodgkin’s lymphoma patients [40].

Here we have assessed the expression of various HSP families, including cytosolic (HSP90α, HSP90β, HSP70, HSC70, and HSP40/DNAJB1), endoplasmic reticulum (GRP78), and mitochondrial (HSP60) HSP homologs in resting and CD3/CD28-stimulated T cells in healthy controls and B-cell lymphoma patients. It should be noted that HSP90α, HSP70, GRP78, and HSP40/DNAJB1 are stress inducible, while HSP90β, HSC70, and HSP60 are constitutively expressed HSP homologs [4]. We found that the expression of HSP90α, HSP90β, HSP70, HSC70, GRP78, HSP40, and HSP60 increases in response to CD3/CD28 stimulation. These results are consistent with previous findings where it was found that phytohemagglutinin (PHA) and IL-2 upregulate HSP70 and HSP90 expression in T cells [45]. We found that BCL patients have dysregulated expression of intracellular and extracellular HSPs in CD3/CD28-stimulated T cells. These data suggest that the upregulation of cytosolic, ER, and mitochondrial HSP homologs is a normal HSP response to T cell activation and that this response is dysregulated in B-cell non-Hodgkin’s lymphoma patients.

It is also interesting to note that the expression of extracellular HSPs did not change following 48 h of CD3/CD28 stimulation in T cells of healthy controls, whereas BCL patients showed aberrant expression of HSPs on the surface of CD3/CD28-activated T cells (Figure 1C; Appendix A), though the results did not reach statistical significance (*p* > 0.05). Further studies should explore the expression of extracellular HSP homologs for longer T cell activation times (i.e., 72–96 h). Targeting eHSPs on dysregulated T cells may be a promising strategy for the treatment of lymphoma patients.

Taking into account that HSPs and immune checkpoints are associated with T cell activation and that current and emerging treatments of lymphoma patients include immune checkpoint inhibitors, we have also assessed the effect of a combined blockade of HSP90 and immune checkpoints (PD-1 and CTLA-4) in vitro. Firstly, we showed that HSP90 inhibition downregulates CD4 and CD8 on the surface of T cells in healthy controls and BCL patients. These results are consistent with previous findings showing that the HSP90 inhibitor downregulates CD4 and CD8 surface expression [41]. We further showed that HSP90 inhibition downregulated CD107a on the surface of activated T cells in healthy controls, suggesting that normal T cells should decrease CD107a on their surface in response to HSP90 blockade. However, CD107a expression on T cells of BCL patients was not affected by the HSP90 inhibitor, suggesting that HSP response to T cell activation is dysregulated in BCL patients. Moreover, immune checkpoint blockade did not abrogate this effect in CD4^+^ and CD8^+^ T cells in BCL patients. It is also interesting to note that stimulation with allogeneic mo-DCs and CD3/CD28 showed a slightly different CD107a expression pattern in response to HSP90 and immune checkpoint inhibition in CD8^+^ T cells, suggesting that the role of combinatorial treatment should be assessed using different T cell stimulations. In this regard, Trojandt et al. reported that the HSP90 inhibitor may affect mo-DC responses, suggesting that even though mixed lymphocyte reactions are effective in exploring the effect of immune checkpoint blockade on T cells, the immune modulatory functions of HSPs should also be taken into account when studying combination therapies [46].

We have also analyzed the effect of HSP90 inhibition on the intracellular expression of PD-1 and CTLA-4 in CD3/CD28-stimulated T cells. The HSP90 inhibitor did not significantly affect the level of immune checkpoints in activated T cells. These data suggest that HSP90, PD-1, and CTLA-4 may function through different mechanisms during T cell activation, however, this requires more investigation.

The study has some limitations. One of the major limitations is the small number of participants, as the study involved only newly diagnosed patients with specific subtypes of lymphoma and with no previous history of treatment.

It should also be noted that the study only explored co-inhibitory immune checkpoints (PD-1 and CTLA-4). Further studies should explore this combinational therapy by targeting co-stimulatory immune checkpoints, including glucocorticoid-induced TNF receptor family-related protein (GITR), OX40, 4-1BB, and inducible T cell co-stimulator (ICOS). Targeting specific HSP homolog and co-inhibitory/co-stimulatory immune checkpoints on T cells may be a promising therapeutic strategy for cancer patients, however, this requires further investigation.

## 5. Conclusions

Heat shock proteins act as molecular chaperones, ensuring cell survival and integrity. Recent studies have highlighted the important role of intracellular and extracellular HSPs in the regulation of immune cells. However, the roles of cytosolic, mitochondrial, and endoplasmic reticulum HSP homologs in T cell activation are largely unknown. Here we show that various HSPs, including HSP90, HSP70, HSP60, GRP78, and HSP40, are upregulated in CD3/CD28-stimulated T cells. Patients with B-cell lymphoma were found to have a dysregulated HSP response to CD3/CD28 stimulation. HSP90 inhibition combined with PD-1 or CTLA-4 inhibitors differently affected CD4+ and CD8+ T cell degranulation responses in activated T cells in BCL patients. Additionally, we found that HSP90 inhibition does not affect intracellular PD-1 and CTLA-4 expression during T cell activation. These studies might help in the identification of new therapeutic targets for the treatment of cancer patients and improve our understanding of HSP functions in immune cells.

## Figures and Tables

**Figure 1 biomedicines-10-02747-f001:**
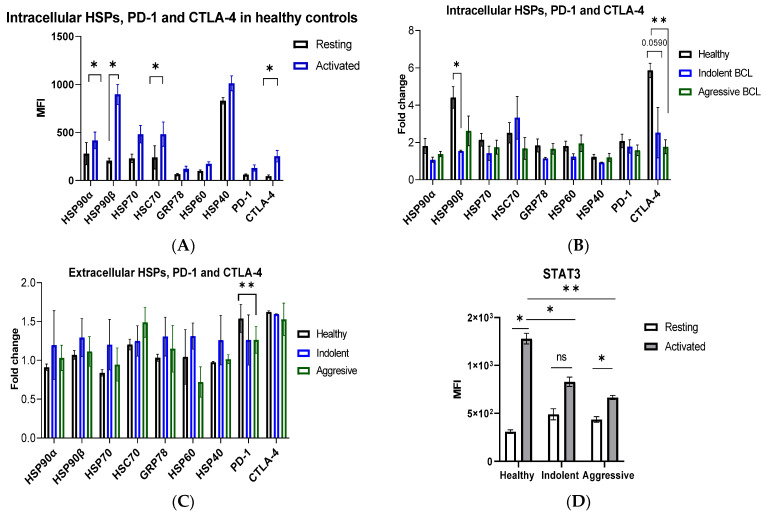
HSP, PD-1, CTLA-4, and STAT3 expression in activated T cells. (**A**) Intracellular expression of HSPs, PD-1, and CTLA-4 (as mean fluorescence intensity, MFI) in activated T cells in healthy controls. Pan T cells were isolated from healthy PBMCs and stimulated with anti-CD3/CD28 or unstimulated (resting) for 48 h. (**B**) Intracellular HSPs, PD-1, and CTLA-4 in indolent and aggressive lymphoma patients. (**C**) Extracellular HSPs, PD-1, and CTLA-4 in indolent and aggressive lymphoma patients. The fold change in MFI was calculated by normalizing the activated to the resting T cells. (**D**) STAT3 expression in resting and activated T cells in healthy controls and lymphoma patients. Data represents mean ± SEM. ns, not significant, * *p* < 0.05, ** *p* < 0.01.

**Figure 2 biomedicines-10-02747-f002:**
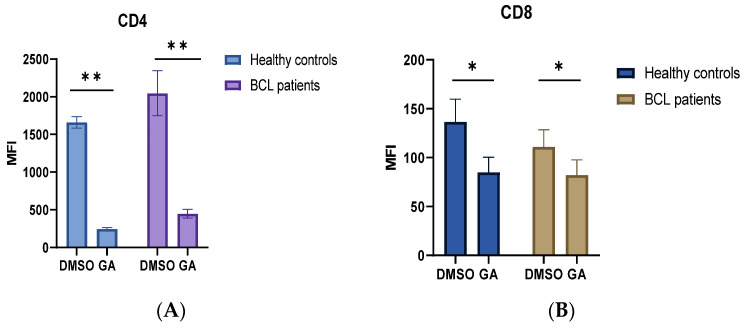
HSP90 inhibition downregulates CD4 and CD8 surface expression. HSP90 inhibitor downregulates surface expression of CD4 (**A**) and CD8 (**B**). T cells were incubated for 4 days with allogeneic mo-DCs in the presence or absence of 0.1 µM geldanamycin (GA). (**C**) Representative flow cytometry plots showing downregulation of CD4 and CD8 molecules on the surface of T cells in response to GA in MLR. Data are presented as mean ± SEM. * *p* < 0.05, ** *p* < 0.01.

**Figure 3 biomedicines-10-02747-f003:**
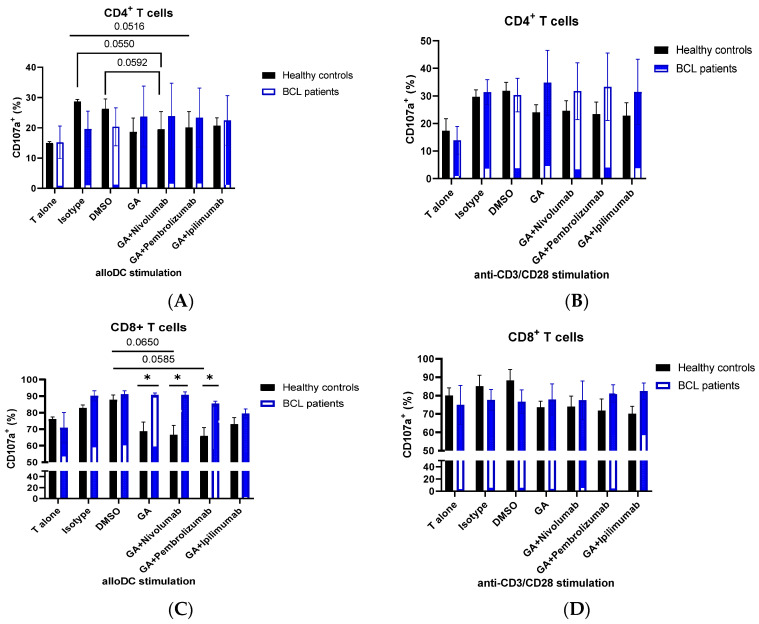
HSP90 inhibition affects CD4^+^ and CD8^+^ T-cell degranulation responses. Degranulation response, as measured by the surface expression of CD107a on CD4^+^ T cells (**A**,**B**) and CD8+ T cells (**C**,**D**) in healthy controls and BCL patients. Isolated T cells were stimulated with allogeneic mo-DCs (**A**,**C**) or anti-CD3/CD28 (**B**,**D**) in the presence or absence of 0.1 µM geldanamycin, 10 µg/mL PD-1 inhibitor (Pembrolizumab/Nivolumab), 10 µg/mL CTLA-4 inhibitor (Ipilimumab). (**E**) Representative flow cytometry plots showing a decrease in CD107a expression on CD4^+^ and CD8^+^ T cells in healthy controls following HSP90 inhibition. Data are presented as mean ± SEM. * *p* < 0.05.

**Figure 4 biomedicines-10-02747-f004:**
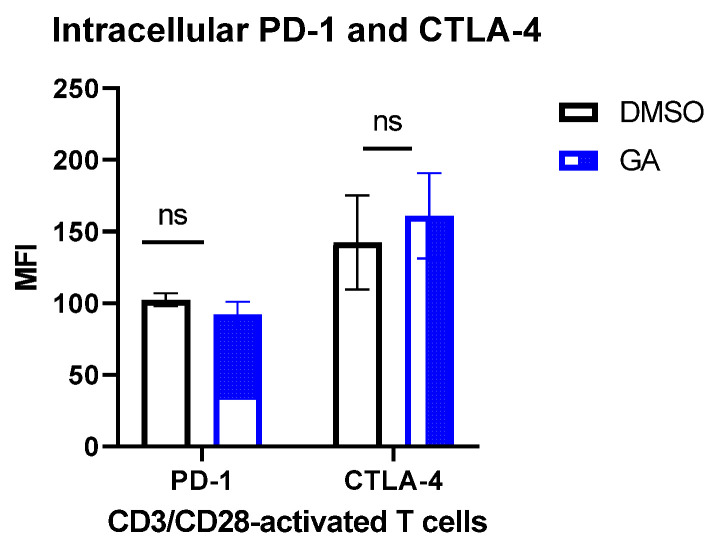
HSP90 inhibition and intracellular PD-1 and CTLA-4 in CD3/CD28-activated T cells. T cells were stimulated for 48 h with anti-CD3/CD28 and hIL-2. At 48 h of incubation, 0.1 µM GA or DMSO was added, and the cells were incubated for an additional 24 h. The graph shows the mean ± SEM. ns, not significant. DMSO, dimethyl sulfoxide; GA, geldanamycin.

## Data Availability

The original contributions presented in the study are included in the article/Appendix A. Further inquiries can be directed to the corresponding author.

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
