# Peer review of "Impaired Heat Shock Protein Expression in Activated T Cells in B-Cell Lymphoma"

_biomedicines, 2022, doi:10.3390/biomedicines10112747_

Round 1

Reviewer 1 Report

The authors present a study on HSPs in T cells and show that HSPs are upregulated following CD3/CD28 stimulation. They show that B-cell lymphoma patients have dysregulated expression of intracellular and extracellular HSPs, immune checkpoints (PD-1, CTLA-4) and STAT3 in CD3/CD28-activated T cells. They further provide evidence that HSP90 inhibition downregulates CD4 and CD8 markers in healthy controls and BCL patients, and differentially affects CD4+ and CD8+ T-cell degranulation response when stimulated with allogeneic DCs or CD3/CD28 in BCL patients (the effect of anti-immunecheckpoint antibodies together with HSP90 inhibitor GA was also examined). Nevertheless, HSP90 inhibition did not significantly affect intracellular PD-1 and CTLA-4 expression in CD3/CD28-activated T cells.

The data is systematically presented, together with statistical analysis of differences. The article would be much improved if the authors include the comments on the topic they touch briefly in the concluding paragraphs: how would they envision the newly identified therapeutic targets for the treatment of cancer patients? What would their proposed therapeutic be (antibody, bispecific, small molecule, combination)? This would increase the translational value of the otherwise excellent contribution.

Please find below a list of minor comments, which I hope you will find helpful.

Line 87: provider missing

Line 99: source of IgG4 isotype

Lines 103-104: concentrations missing

Line 110: concentration of DMSO

Line 128: Figure 1D appears in text before 1C

Line 156: HSP: something wrong with bold font and the punctuation?

Line 167: T-cell degranulation response (and throughout the text for expressions related to T-cell properties)

Line 176: did not abrogate

Figure 2: Panels are not clearly labelled, it is not clear which plot corresponds to which subpanel.

Line 203: Plot 3A is skewed. It is not clear what the flow cytometry histogram represents (I would include 3F subpanel).

Line 237: please reword: where it was found that PHA and IL-2 upregulate…

Line 245: The effect on CD4 and CD8 of anti-immunecheckpoint antibodies alone should be presented, especially because they were used at high concentrations

Line 260: mixed lymphocyte reaction?

Line 267: this requires mor investigation (or similar)

Line 277: the sentence „Additionally, we found that healthy controls and BCL patients have dysregulated STAT3, PD-1 and CTLA-4 expression“ does not make sense, please amend

Author Response

Dear Sir/Madam,

We truly thank you for your comments.

We have added the information on the topic in the Discussion section (Lines 297-304).

Lines 100,101,113, 117: We have added the information on provider/source and concentrations.

Line 163: Figures 1D and 1E were moved to Supplementary materials, since the information from these figures is included in Figures 1B and 1C.

Line 167: bold font and punctuation were modified during word to pdf conversion. We have corrected font and punctuation.

Line 175: We have changed T cell degranulation response to T-cell degranulation response throughout the text

Line 188: we have changed “ did not abrogated” to “ did not abrogate”

Figure 2 and Line 205, 209, Figure 3, Line 218, 225: We have labeled the figures to make them clearer

Line 253: we have changed “ showing that..” to “ where it was found that“

Line 271:  The concentration 10ug/ml for anti-PD1 and anti-CTLA-4 antibodies was reported in previous studies. The main aim of our experiment was to examine the effect of combined blockade of HSP90 and immune ckeckpoints on T cells.

Line 285: we have changed to “ mixed lymphocyte reaction”

Line 293: We have changed to “requires more investigation”

Line 315: we have rephrased the sentence

Reviewer 2 Report

Albakova et al. have investigated the function of heat shock protein expression in lymphoma in this manuscript. Even though this is an interesting work, it must be improved for clarity and continuity.

  1. The authors should elaborate on Heat shock proteins, their structural features, and their biological significance. Refer “Heat shock proteins (HSPs) are a family of chaperones contributing to the protein 27 homeostasis and survival of the cell”
  2. The authors must elaborate on the parameters utilized for sample size calculation, study power calculation, and effect size calculation.
  3. The authors must specify the housekeeping genes utilized in this investigation. How were the fold changes in the manuscript identified? Additionally, the authors should share the raw data for reproducibility.
  4. Authors can go into more detail by considering the roles that extracellular HSPs play. “Originally 31 described as intracellular proteins, several studies reported the presence of HSPs in the 32 plasma/serum of cancer patients, in extracellular vesicles and on the surface of tumor 33 cells, collectively called extracellular HSPs”

Author Response

Dear Sir/Madam,

Thank you for your comments. We truly appreciate them.

  1. We have added information on the functions of HSPs in Introduction (Lines 28-29, 33-35).
  2. We have added information, concerning sample size in Discussion section; we have added information that small sample size is the limitation of our study (Lines 294-296).
  3. In this study, we have not assessed gene expression. The main aim of this study was to assess HSP protein expression. We have provided information on how fold change was calculated (line 172). 
  4. We have added information on extracellular HSPs (Lines 37-41).

Reviewer 3 Report

In the manuscript with the title “Impaired Heat Shock Protein Expression in Activated T cells in Lymphoma”, the authors assess the expression of intracellular and extracellular heat-shock proteins (HSPs), PD-1, CTLA-4, and STAT3 in CD3/CD28-activated and non-activated T cells deriving from healthy blood donors and patients with B cell lymphomas. Even though the findings of the study seem significant, several major issues exist, as explained below:

Major issues:

·         The aim of the study is not clearly presented and therefore I cannot properly assess if it was achieved.

·         T cells were isolated from B cell lymphoma patients and then activated. Therefore, as this is an in vivo experimental procedure it is advised to show this fact in all parts of the manuscript, for example in the title.

·         The number of samples is extremely limited (≤5 in each category). It is advised to include samples from more patients in order to present results with certainty in statistical significance.

·         Background information is extremely limited. For example, what is the role of PD-1, CTLA-4, and STAT3 in T cells? This is a major issue as the authors skip some essential information for the comprehension of the manuscript.

·         The Materials and Methods section is extremely confusing. The experimental protocol is not clear after the isolation of PBMCs. T cells were activated once? The times of the treatments are not clearly presented. It is advised to present in detail every step of this section as it is extremely important for a potential repetition of the study.

·         The authors state that: “These data suggest that HSP upregulation is a normal cellular response to T cell activation and that this response is dysregulated in T cells in lymphoma patients, which was further supported by the dysregulated expression of immune checkpoints and STAT3 in lymphoma patients”. However, this observation seems to happen only in the intracellular expression of HSPs. Moreover, the authors do clarify if an observation refers to activated or non-activated T cells.

·         The Discussion section is extremely limited and does not properly integrate the results of the study into the relative scientific knowledge.

·         What is MFI and why do the authors provide it in the axis of some figures? What is the difference with fold change which is illustrated in other figures?

·         The figures need to be enlarged as their current size seems small.

·         Figures 1a, 1b, and 1c seem to be included in figures 1d and 1e. Therefore, it is advised to exclude them.

Minor issues:

·         It is advised to state an inhibitor as a PD-1 inhibitor and not with its common name. This would greatly improve the clarity of the manuscript.

·         BCL in the Abstract section: It is advised to explain each abbreviation the first time that it is stated.

·         What is MLR?

·         The word lymphomas is included in the BCL.

·         It is advised to change the word “client”.

Author Response

Dear Sir/Madam,

Thank you for your comments. We truly appreciate them.

We have added the aim of the study in the Introduction section (Lines 61-64).

We have added information that these experiments were performed in vitro (Lines 64, 176,271).

We have added information that small sample size is the limitation of our study (Line 294).

We have stimulated T cells once. All experimental procedures were performed according to the manufacturer instruction.

We have added information that the observation was made on intracellular HSPs in activated T cells (Lines 156, Line 157). We have also added the observation made on extracellular HSPs (Lines 159-162).

We have added information on the results in the Discussion section (Lines 260-267, 297)

We have added the information on MFI and fold change ( lines 169, 172).

We have enlarged the figures.

We have moved Figure 1D  and 1E to the Supplementary materials.

We have changed common names of immune checkpoint inhibitors to PD-1 or CTLA-4 inhibitors.

We have explained all the abbreviation, including BCL and MLR.

We used the term "client", since it is commonly used term for proteins that are processed by HSPs.

Round 2

Reviewer 2 Report

The authors have improved the manuscript but the sample size is the major limitation of this manuscript. 

Author Response

Dear Sir/Madam,

Thank you for your comments. We truly appreciate them. During the 1st round of revision, we have added the section in the Discussion emphasizing that the study involved small number of participants because only newly diagnosed/primary patients with specific types of Non-Hodgkin B-cell lymphoma and with no previous history of treatment were included in the study. These patients are unique and rare patients in hematology clinics. However, since rare patients were involved in the study, the findings of this study offer new insights on the role of HSPs in T cell activation and may be potentially useful for the discovery of novel immunological targets for the treatment of newly diagnosed B-cell lymphoma patients.

Reviewer 3 Report

The manuscript with the title “Impaired Heat Shock Protein Expression in Activated T cells in Lymphoma” has been improved after the first round of revision. However, a few minor issues still exist as described below:

·         T cells were isolated from B cell lymphoma patients and then activated. Therefore, a rephrasing of the title to “… in B-cell lymphoma” is strongly recommended.

·         Text in figures 1 and 3 is not clear. It is advised to enlarge it.

·         It is advised to refer to inhibitors with consistency in all parts of the manuscript and figures.

Author Response

Dear Sir/Madam,

Thank you for your comments.

We have rephrased the title by adding “B-cell lymphoma”.

We have enlarged the text for all figures.

We have changed “anti-PD-1” or “anti-CTLA-4” to “PD-1 inhibitor “ and  “CTLA-4 inhibitor”, respectively, in all parts of the manuscript and figures.